Relative availability of natural prey versus livestock predicts landscape suitability for cheetahs Acinonyx jubatus in Botswana

Winterbach Hanlie E.K. 1 2 tauconsultants@gmail.com
Winterbach Christiaan W. 1 2
Boast Lorraine K. 3
Klein Rebecca 3
Somers Michael J. 2 4
1 Tau Consultants (Pty) Ltd , Maun , Botswana
2 Centre for Wildlife Management, University of Pretoria , Pretoria , South Africa
3 Cheetah Conservation Botswana , Gaborone , Botswana
4 Centre for Invasion Biology, University of Pretoria , Pretoria , South Africa
Varela Sara
Electronic publication date: 2015 Jul 9
Publication date: 2015
Volume: 3
Electronic Location ID: e1033
Received 2015 Feb 5; Accepted 2015 May 27
Copyright: © 2015 Winterbach et al.
Copyright year: 2015
Copyright holder: Winterbach et al.
License: This is an open access article distributed under the terms of the Creative Commons Attribution License, which permits unrestricted use, distribution, reproduction and adaptation in any medium and for any purpose provided that it is properly attributed. For attribution, the original author(s), title, publication source (PeerJ) and either DOI or URL of the article must be cited.
License URL: https://creativecommons.org/licenses/by/4.0/

Keywords: Cheetah, Landscape suitability, Prey availability, Human-carnivore conflict, Botswana

Funding: This work was partly funded by Tau Consultants (Pty) Ltd. The funders had no role in study design, data collection and analysis, decision to publish, or preparation of the manuscript.

==============================
Prey availability and human-carnivore conflict are strong determinants that govern the spatial distribution and abundance of large carnivore species and determine the suitability of areas for their conservation. For wide-ranging large carnivores such as cheetahs (Acinonyx jubatus), additional conservation areas beyond protected area boundaries are crucial to effectively conserve them both inside and outside protected areas. Although cheetahs prefer preying on wild prey, they also cause conflict with people by predating on especially small livestock. We investigated whether the distribution of cheetahs’ preferred prey and small livestock biomass could be used to explore the potential suitability of agricultural areas in Botswana for the long-term persistence of its cheetah population. We found it gave a good point of departure for identifying priority areas for land management, the threat to connectivity between cheetah populations, and areas where the reduction and mitigation of human-cheetah conflict is critical. Our analysis showed the existence of a wide prey base for cheetahs across large parts of Botswana’s agricultural areas, which provide additional large areas with high conservation potential. Twenty percent of wild prey biomass appears to be the critical point to distinguish between high and low probable levels of human-cheetah conflict. We identified focal areas in the agricultural zones where restoring wild prey numbers in concurrence with effective human-cheetah conflict mitigation efforts are the most immediate conservation strategies needed to maintain Botswana’s still large and contiguous cheetah population.

Introduction

The cheetah is Africa’s most endangered felid (Marker et al., 2007) and is listed as Vulnerable with a declining population trend in the IUCN Red List of Threatened Species (Durant et al., 2008). Botswana is important for the regional and global long-term survival of cheetahs. It has a large and still contiguous cheetah population, and hosts the second largest national population with ±1,786 animals (Klein, 2007) after Namibia with ±3,138–5,775 animals (Hanssen & Stander, 2004; Marker et al., 2007). It also forms the major connecting range for the southern African cheetah population, which is the largest known free-ranging resident population comprising ±6,500 animals (IUCN/SCC, 2007).

At least 75% of the cheetah’s resident range in southern Africa falls outside protected areas (IUCN/SCC, 2007), mostly on farmlands where competition with other large carnivores is low and a sufficient small to medium-sized wild prey base still occurs (Marker & Dickman, 2004; Klein, 2007; Lindsey & Davies-Mostert, 2009). Cheetahs are one of the most wide-ranging terrestrial carnivores and need extensive areas to sustain viable populations (IUCN/SCC, 2007; Belbachir et al., 2015). Interspecific competition, especially from lions (Panthera leo) and spotted hyaenas (Crocuta crocuta), is suggested as one of the main reason for this wide-ranging behaviour as it can strongly influence their movements, behaviour and density (Durant, 2000; Broekhuis et al., 2013). Competition between cheetahs and their dominant competitors occurs predominantly inside protected areas, where densities of lions and spotted hyaenas tend to be high (Creel, Spong & Creel, 2001). With few protected areas large enough to contain the ranging behaviour of cheetahs, the provision of additional areas beyond protected area boundaries is necessary to effectively conserve them both inside and outside protected areas (Woodroffe & Ginsberg, 1998). Consequently, conservation efforts in human-dominated landscapes are critical for the cheetah’s long-term survival.

The two primary factors influencing large carnivore survival is prey availability and conflict with humans (Winterbach et al., 2013). The strong linear relationships that exist between the density of African large carnivores and the biomass of their natural prey (Hayward, O’Brien & Kerley, 2007) point to prey availability as the primary natural determinant that governs the spatial distribution and abundance of large carnivore species, and hence the suitability of an area for their conservation (Gittleman & Harvey, 1982; Fuller & Sievert, 2001; Hayward, O’Brien & Kerley, 2007; Broekhuis et al., 2013). However, although cheetahs prefer wild prey (Marker et al., 2003), they also cause conflict with people by preying on livestock, usually small stock (sheep and goats) and occasionally calves and foals (Marker et al., 2003; Ogada et al., 2003; Woodroffe et al., 2007; Selebatso, Moe & Swenson, 2008). As a result, in human-dominated landscapes, human activities and conflicts with predators often become as strong a determining factor as prey availability in the existence of large carnivores, including cheetahs (Woodroffe & Ginsberg, 1998; Marker et al., 2003; Gusset et al., 2009; Schuette, Creel & Christianson, 2013).

In 2009, the Botswana Department of Wildlife and National Parks (DWNP) drafted a national conservation action plan for the country’s cheetahs and African wild dogs. One of the primary targets set out in the action plan is to obtain quantitative knowledge regarding the main threats to securing a viable cheetah population across its range in Botswana. Scientific information on cheetah distribution and density on a country-wide scale, however, is nearly impossible to obtain, but the urgent conservation status of this species means we can ill-afford to wait for detailed scientific information before policy decisions are made. Therefore, an objective, clear system of evaluation, based on the best available and reliable information is needed, that can be used as a basis to support policy setting (Theobald et al., 2000).

In this study we aimed to assess potentially suitable habitat across Botswana for the long-term persistence of cheetahs using wild prey distribution and a proxy for probable human-cheetah conflict. Such an assessment is especially important in the extensive agricultural areas used primarily for raising livestock, where around half of Botswana’s cheetah population occurs (Winterbach & Winterbach, 2003). Data from country-wide aerial survey counts of both wild prey and livestock were the best available information for this assessment. We calculated cheetah prey biomass from the aerial counts of wild species likely to be utilized by cheetahs in Botswana. As a proxy for probable levels of human-cheetah conflict, we calculated the proportion of cheetah prey biomass relative to the combined biomass of wild prey and small livestock. By combining this information with country-wide reports of human-cheetah conflict, we created a predictive map of landscape suitability for cheetahs. This approach to making a landscape assessment for a specific species on a national scale is unique, and generated fundamental knowledge that can inform decisions regarding land management, connectivity between cheetah sub-populations, and human-cheetah conflict mitigation. The predictive map highlighted some important implications for the conservation of cheetahs in Botswana, which we discuss.

Study Area

The Republic of Botswana is ca 582,000 km2 in size and is landlocked with Namibia, South Africa, Zimbabwe, and Zambia as its neighbours. Roughly 50% of its 2 million people (3.5 people/km2) live in rural villages and small settlements (Central Statistics Office, 2014).

The mean altitude above sea level is 1,000 m (515–1,491 m a.s.l.). The climate is arid to semi-arid with highly variable rainfall and severe droughts. Mean annual rainfall varies from 650 mm in the north-east to 250 mm in the south-west. Average maximum daily temperatures range from 22 °C in July to 33 °C in January and average minimum temperatures from 5 °C to 19 °C, respectively (Department of Surveys and Mapping, 2001). The country has only two perennial rivers; the Okavango River which fans out into the Okavango Delta and the Kwando/Linyanti/Chobe river system which forms the boundary with Namibia and Zambia. The Makgadikgadi Pans is a seasonal wetland with natural perennial water holes in the ephemeral Boteti River providing critical dry season water sources for wildlife in Makgadikgadi Pans National Park. Across the rest of the country, scattered pans and ancient riverbeds periodically hold water during the wet season. Considerable seasonal variations in the density and distribution of ungulate species occur and the blocking of migration routes by veterinary fences has led to ungulate die-offs during drought years (Verlinden, 1998; Bergström & Skarpe, 1999). Seasonal migrations of Burchell’s zebra and blue wildebeest (Connochaetes taurinus) still occur inside Makgadikgadi Pans National Park (Brooks, 2005), and zebra migrate between Makgadikgadi Pans National Park and the Okavango Delta (Bartlam-Brooks, Bonyongo & Harris, 2011).

Approximately 38% of the land use in Botswana is designated for wildlife utilization; 17% as protected areas (national parks and game reserves) and 21% as Wildlife Management Areas. The latter are primarily designed for wildlife conservation, utilisation, and management (Hachileka, 2003). The Northern Conservation Zone consists of Chobe, Nxia Pan, and Makgadikgadi Pans National Parks, Moremi Game Reserve in the Okavango Delta, and Wildlife Management Areas. The Central Kalahari Game Reserve and Kgalagadi Transfrontier Park are linked with Wildlife Management Areas to form the Southern Conservation Zone. Protected areas and Wildlife Management Areas do not have ‘predator-proof’ fences, with the exception of the western and southern boundary of Makgadikgadi Pans National Park which provides only a partial barrier due to lack of maintenance.

Five percent of the country is urban, 57% is rangeland (of which roughly 70% is tribal / communal grazing land), 25% is state land, and 5% is freehold land leased for large-scale commercial ranching (Department of Surveys and Mapping, 2001). In the Draft National Predator Strategy (Winterbach & Winterbach, 2003), the country was sub-divided into two main predator management zones: the Conservation Zone comprises national parks, forest reserves, sanctuaries, and Wildlife Management Areas; the Agricultural Zone comprises rangelands, residential, and mining areas (Fig. 1).

Figure 1 Land use categories and predator management zones in Botswana.

Land use categories and predator management zones in Botswana. The predator management zones were identified in Winterbach, Winterbach & Somers (2014). The Conservation Zone (CZ) comprises protected areas (national parks, game and forest reserves, and sanctuaries) and Wildlife Management Areas. The Agricultural Zone (AZ) comprises rangeland, residential, and mining areas.

Livestock (mainly cattle) rearing is the primary economic activity over large parts of Botswana and constitutes 70–80% of the agricultural GDP (Botswana Ministry of Agriculture, 2011). In the 2012 household survey, the livestock population in Botswana was estimated as 2.6 million cattle, 1.8 million goats, and 300,000 sheep, most of which were located on the more fertile eastern side of the country. Approximately 92% of these livestock are farmed traditionally using the cattle post system on communal grazing land. Botswana’s key environmental issues include water scarcity and pollution, rangeland degradation and desertification, loss of biodiversity, deforestation, and an increased frequency of periodic droughts (Wingqvist & Dahlberg, 2008).

Methods

Cheetah wild prey were represented by those animal species counted in aerial surveys across Botswana that are within the accessible prey weight range (body mass 14–135 kg) for cheetahs identified by Clements et al. (2014) (Table 1), hereafter termed ‘cheetah prey’. For small livestock we used goats and sheep (hereafter referred to as ‘small stock’) as the main livestock whose depredation is a significant predictor of human-cheetah conflict levels, as illustrated by problem animal control (PAC) reports from DWNP (Data S1). The source of the estimated number of each prey species and small stock was the Botswana Aerial Survey Information System database (Botswana Government, 2000), which holds the annual dry-season country-wide survey data.

Table 1 Cheetah prey species masses (kg) used in the conversion calculations from aerial survey counts to Large Stock Units (LSU) and the corresponding LSU values used to determine biomass.

Species	Scientific name	Weight (kg)a	LSU conversiona	
Tsessebe	Damaliscus lunatus lunatus	110	0.34764	
Red lechwe	Kobus leche	72	0.25298	
Ostrich	Struthio camelus	68	0.24237	
Warthog	Phacochoerus africanus	45	0.17783	
Impala	Aepyceros melampus	45	0.17783	
Reedbuck	Redunca arundinum	40	0.16279	
Springbok	Antidorcas marsupialis	26	0.11785	
Duiker	Sylvicapra grimmia	15	0.07801	
Steenbok	Raphicerus campestris	10	0.05756	
Notes.

a Source: Botswana Aerial Survey Information System database (Botswana Government, 2000).

We calculated the mean number per cheetah prey species for each 12′ grid cell (approximately 21 × 22 km) for the 2001 to 2005 and 2007 surveys. We excluded the survey conducted in 2006, as it only covered northern Botswana, and those conducted during the drought years before 2001 and very wet years after 2007. This was done to avoid including counts from extreme years when large variations in population estimates is likely. The mean number per species were then converted to species biomass in Large Stock Units (LSU), using the formula LSU = number of animals × body weight0.75 (Meissner, 1982), and summed to provide the total cheetah prey and small stock biomass, respectively, per grid cell. We calculated the percentage that cheetah prey biomass contributed to the total biomass consisting of cheetah prey and small stock (hereafter referred to as ‘percentage cheetah prey’ biomass), where a low percentage indicated high probable levels of conflict (Data S2).

Cheetah biomass strongly correlates with dry season prey biomass (Fuller & Sievert, 2001), and we felt the combined data set of the six aerial surveys used in this study best represented the general distribution of prey and small stock biomass on a country-wide scale. Although aerial surveys tend to undercount small mammal species such as steenbok (Raphicerus campestris) and duiker (Sylvicapra grimmia) (Jachmann, 2002), it is the only feasible method for wildlife monitoring over such large areas. We utilized the existing broad landscape suitability stratification for large carnivores in Botswana from Winterbach, Winterbach & Somers (2014), and refined the agricultural zones to identify homogeneous strata based on the distribution of cheetah prey biomass, small stock biomass, and percentage cheetah prey biomass.

We tested the hypothesis that percentage cheetah prey was correlated with the level of conflict. We used data on livestock attacks by cheetahs between 1995 and 2006 consisting of PAC reports, and farms questionnaire surveys (N = 188) conducted during 2004 and 2005 (Klein, 2007), to determine if the number of livestock attacks changed with the percentage cheetah prey. We used the following intervals of percentage cheetah prey: 0, >0 to ≤1, >1 to ≤2, >2 to ≤5, >5 to ≤10, >10 to ≤20, >20 to ≤80, and >80 to ≤100, and determined the number of grid cells and the number of reported attacks per interval. Then we calculated the number of attacks per grid cell for each interval and the overall mean number of attacks per grid cell.

We used a chi square test to determine if the number of grid cells (expected values) and the number of livestock attacks (observed values) differed significantly for the following categories of percentage cheetah prey: 0, >0 to ≤20, >20 to ≤80, and >80 to ≤100. Observations were classified into categories independently and all categories had expected frequencies >5%. We used Bonferroni intervals (Byers, Steinhorst & Krausman, 1984) to test for categories with observed frequencies that differed significantly from the expected. Based on the results, we identified the threshold of percentage cheetah prey to differentiate between low and high probable levels of conflict. On the premise that strata with high probable levels of conflict will have a low suitability for the long-term survival of cheetahs, we calculated for each stratum the percentage grid cells with ≤20% cheetah prey. We then used the Natural Breaks (Jenks) function in ArcMap 9.3.1 to classify the strata into five classes of relative landscape suitability. The Natural Breaks (Jenks) function grouped similar values and maximized the differences between classes.

We used presence and absence data of cheetahs to test our landscape classification of strata as suitable (very high to low suitability combined) or unsuitable. Game ranchers and commercial livestock farmers in the game ranching regions of the Central, Ghanzi, Ngamiland and North East regional districts (Boast, 2014) reported the status of cheetahs on their properties in questionnaire surveys (N = 89), conducted during 2012 and 2013. Cheetahs were reported as present (visual sightings or tracks seen at least quarterly), transient (visual sightings or tracks seen less frequently than quarterly), or absent (never seen cheetahs or its tracks). We used a chi-square test with Bonferroni simultaneous confidence intervals (Byers, Steinhorst & Krausman, 1984) to test the hypothesis that farmers reported cheetahs as present or absent in similar proportions to the number of grid cells per suitable and unsuitable landscape classes.

Results

Cheetah prey occurred widely across Botswana, with the highest biomass in the Okavango Delta and Kwando/Linyanti/Chobe river system in the Northern Conservation Zone and in the Tuli Conservation Zone in eastern Botswana (Fig. 2). Cheetah prey biomass was low across most part of the Ngamiland Agricultural Zone in the north-west. Small stock occurred widely across the agricultural zones, but was unevenly distributed (Fig. 3). The percentage cheetah prey differed markedly between the strata identified in the agricultural zones (Fig. 4). A list of strata names and identification numbers are supplied in Data S3. In the eastern parts of the Central Agricultural Zone (stratum 5.3), and in the Kgalagadi Agricultural Zone 1 (stratum 8.3), the Makgadikgadi Pans Community stratum (stratum 5.6), and the two strata surrounding the Okavango Delta (strata 6.2 and 6.4), cheetah prey contributed only 0–5% of the total biomass (cheetah prey plus small stock). This indicates very high probable levels of conflict in these areas.

Figure 2 The distribution of cheetah prey biomass in Large Stock Units (LSU) across Botswana.

Cheetah prey are represented by those wild animal species counted in six aerial surveys across Botswana (2001 to 2005, and 2007) that are within the accessible prey weight range (body mass 14–135 kg) for cheetahs, identified by Clements et al. (2014). The stratification of the Conservation and Agricultural Zones into strata is identified by number (Data S3).

Figure 3 The distribution of small stock biomass in Large Stock Units (LSU) across Botswana.

Small stock is represented by goats and sheep as the main livestock whose depredation is a significant predictor of human-cheetah conflict levels in Botswana. This was illustrated by problem animal control (PAC) reports (Data S1). The stratification of the Conservation and Agricultural Zones into strata is identified by number (Data S3).

Figure 4 The distribution of the percentage cheetah prey biomass in Large Stock Units (LSU).

Percentage cheetah prey biomass is the proportion of cheetah prey biomass relative to the total biomass (cheetah prey and small stock biomass combined) and was used as a proxy for probable levels of human-cheetah conflict. A low percentage indicated high probable levels of conflict. The stratification of the Conservation and Agricultural Zones into strata is identified by number (Data S3).

Livestock attacks recorded per grid cell for the percentage cheetah prey biomass intervals ranged from 0.23 to 0.81 attacks per grid cell with a mean of 0.49 and standard error of 0.25 (N = 8) (Table 2). The number of attacks was consistently below the mean when the percentage cheetah prey biomass exceeded 20%. The observed frequency of attacks between the categories of percentage cheetah prey differed significantly from the expected values (χ2 = 52.42, df = 1, P < 0.001) (Table 3). Livestock attacks were more than expected (P = 0.05) in grids with 0% cheetah prey biomass, and significantly lower than expected in areas with >20% (α = 0.05, Z = 2.4977). We therefore took grids with ≤20% cheetah prey biomass to represent areas with high probable levels of conflict, and >20% cheetah prey biomass to represent areas with low probable levels of conflict.

Table 2 The number of livestock attacks by cheetahs per 12′ aerial survey grid in the different percentage cheetah prey intervals, and the overall mean number of attacks per grid.

The number of attacks was consistently below the mean when the percentage cheetah prey biomass exceeded 20%. Livestock attacks occurred in the Kgalagadi and Ghanzi Agricultural Zones and the western strata of the Central Agricultural Zone between 1995 and 2005 (Klein, 2007).

Percentage cheetah prey	Number of grids	Number of livestock attacks	Attacks per grid	
0	43	34	0.79	
>0 to ≤1	48	39	0.81	
>1 to ≤2	33	9	0.27	
>2 to ≤5	48	27	0.56	
>5 to ≤10	40	9	0.23	
>10 to ≤20	23	15	0.65	
>20 to ≤80	32	8	0.25	
>80 to ≤100	136	47	0.35	
TOTAL	403	188	0.49 (mean)	

Table 3 Bonferroni intervals to test for categories percentage cheetah prey with observed frequencies of livestock attacks by cheetahs that differed significantly from the expected (k = 4, α = 0.05, Z = 2.4977).

A chi square test was used to determine if the number of grid cells (N = 403) (expected values) and the number of livestock attacks by cheetahs (N = 188) recorded (observed values) differed significantly for the four categories of percentage cheetah prey (χ2 = 52.42, df = 1, P < 0.001). From these results, 20% cheetah prey appears to be the critical point to differentiate between low and high probable levels of human-cheetah conflict.

Percentage cheetah prey	Expected proportion Pio	Observed proportion Pi	Bonferonni intervals for Pi	Use index Pi/Pio	Significant	
0	0.106700	0.180851	0.1107 ≤Pi ≤0.2510	1.69	+	
>0 to ≤20	0.476427	0.526596	0.4357 ≤Pi ≤0.6175	1.11	0	
>20 to ≤80	0.079404	0.042553	0.0058 ≤Pi ≤0.0793	0.54	−	
>80 to ≤100	0.337469	0.250000	0.1711 ≤Pi ≤0.3289	0.74	−	

Based on the percentage grid cells per stratum with ≤20% cheetah prey, we rated the five classes of landscape suitability identified in ArcMap as very high (0–6.7% grids), high (6.8–25% grids), medium (25.1–50% grids), low (50.1–75% grids) or unsuitable (75.1–100% grids) (Fig. 5). Data S4 summarises the number of grid cells per stratum, the number and percentage of grid cells with ≤20% cheetah prey, and the suitability class. The classification of some strata as unsuitable for cheetahs was supported by the questionnaire data where the proportion of farmers that reported cheetahs present or absent differed significantly between suitable and unsuitable areas (χ2 = 129.11, df = 3, P < 0.001) (Table 4). Farmers reported cheetahs absent in the unsuitable areas (n = 10) significantly more than would be expected by chance, and only 13.9% of farmers reported cheetahs absent within the suitable areas (N = 79) (α = 0.05, Z = 2.4977).

Figure 5 Predictive map showing the landscape suitability of the different strata in Botswana for the long-term persistence of cheetahs.

Suitability was assessed from the percentage grid cells per stratum with ≤20% cheetah prey biomass and the Natural Breaks (Jenks) function in ArcMap 9.3.1, to classify the strata into five classes of suitability. The stratification of the Conservation and Agricultural Zones into strata is identified by number (Data S3).

Table 4 Bonferonni intervals for the presence, transience, and absence of cheetahs in areas deemed suitable and unsuitable for cheetahs in Botswana (k = 4, α = 0.05, Z = 2.4977).

The proportion of farmers that reported cheetahs present or absent differed significantly between suitable and unsuitable areas (χ2 = 129.11, df = 3, P < 0.001). Data were based on questionnaires completed by game ranchers and commercial livestock farmers (N = 89) during 2012 and 2013 in the game ranching regions of the Central, Ghanzi, Ngamiland and North East regional districts (Boast, 2014). Simultaneous confidence intervals for the presence, transience, and absence of cheetahs based on questionnaires completed by farmers (N = 89) in areas deemed suitable and unsuitable for cheetahs in Botswana (k = 4, α = 0.05, Z = 2.4977).

Observation type	Expected proportion Pio	Observed proportion Pi	Bonferonni intervals for Pi	Use index Pi/Pio	Significant	
Absent in unsuitable area	0.026462	0.112360	0.0287 ≤ Pi ≤0.1960	4.25	*+	
Present in unsuitable area	0.085687	0.000000	0.0000 ≤ Pi ≤ 0.0000	0.00	∗ −	
Absent in suitable area	0.209493	0.123596	0.0365 ≤ Pi ≤ 0.2107	0.59	0	
Present in suitable area	0.678358	0.764045	0.6516 ≤ Pi ≤ 0.8765	1.13	0	

Discussion

Our analysis and the predictive suitability map presented provide a means for assessing the landscape suitability within Botswana for cheetahs. We identified three important features of the predictive map: land management in the agricultural areas to support a sufficient prey base for cheetahs, maintaining connectivity between cheetah sub-populations within Botswana and across its borders, and focal areas for the reduction of human-cheetah conflict.

The aerial survey data show that there is a good wild prey base for cheetahs across large parts of the agricultural areas (Fig. 2) despite the extensive distribution of small stock (Fig. 3). The high density of steenbok and duiker (range 0.261–4.319 animals/100 km2) in the tribal grazing land of the Ghanzi Community Stratum (stratum 7.2) calculated from the six aerial surveys suggests that large and small livestock do not necessarily displace small ungulates to the extent that large ungulates are displaced by cattle (Riginos et al., 2012). This greater resource availability may be causal to the considerably higher density, smaller home range sizes, and generally larger body size of cheetahs in Botswana compared to Namibia (Boast et al., 2013).

The significant reduction in livestock attacks in areas with >20% cheetah prey biomass showed that percentage cheetah prey biomass can be used to distinguish between high and low levels of probable conflict on a landscape level. However, 20% cheetah prey as the likely critical point to distinguish between high and low probable levels of human-cheetah conflict is estimated for Botswana and not necessarily valid for other landscapes. Furthermore, smaller-scale studies using other methods to count wildlife (e.g., transect counts) should be conducted within Botswana, to test whether the critical point we report here remains valid at finer scales of investigation. The fact that livestock attacks by cheetahs decreased significantly with an increase in percentage cheetah prey confirms the findings in Namibia and in Botswana that cheetahs commonly prefer wild prey (Marker et al., 2003; Boast, 2014). A preference for wild prey has also been shown for other large carnivores in areas where livestock is predominant (Hemson, 2003; Woodroffe et al., 2005; Ogara et al., 2010). This trend would likely be supported by efforts to maintain wild prey populations within livestock areas to decrease livestock depredation (Mizutani, 1999; de Azevedo & Murray, 2007). It is, however, important to note that the actual occurrence of conflict is not uniformly spread across areas (Woodroffe, Thirgood & Rabinowitz, 2005; Holmern, Nyahongo & Røskaft, 2007; Kushnir et al., 2010), and is determined by a more complex array of factors than prey availability alone (Winterbach et al., 2013). Thus human-wildlife conflict must also be assessed on a local level scale to guide mitigation strategies (Muntifering et al., 2006).

The landscape suitability map (Fig. 5) shows a number of important strata where the long-term persistence of cheetahs is currently uncertain, and which may have a negative effect on connectivity within the Botswana and for the regional southern African population. In the Northern Conservation Zone, two strata surrounding the Okavango Delta (strata 6.2 and 6.4) were ranked as unsuitable. This could hinder the free movement of cheetahs and thus reduce the connectivity between Botswana’s northern and southern sub-populations. In the south, a large part of the Kgalagadi Agricultural Zone 2 was ranked as currently unsuitable or with a low suitability for cheetahs due to a low percentage prey. In addition, the increasing demand for livestock grazing areas has led to a proposed change in land use from wildlife to cattle east of the Ghanzi Community Stratum (stratum 7.2) (Botswana Ministry of Lands and Housing, 2008). With extensive areas inside the Southern Conservation Zone already used for cattle production (Conservation International Botswana, 2010), a barrier to large carnivore and wildlife movement in general may well develop between Botswana’s two largest protected areas: the Central Kalahari Game Reserve and the Kgalagadi Transfrontier Park. Both these parks have a high irreplaceability value in maintaining connectivity for large carnivores and ungulates in Africa (Wegmann et al., 2014). Almost half of the country’s cheetah population occurs in the Southern Conservation Zone. To safeguard the large and yet contiguous cheetah population in Botswana, where widespread natural movements still allow substantial gene flow (Dalton et al., 2013), conservation planning will have to address ways to secure effective wildlife corridors.

Linkage of cheetah populations between Botswana and neighbouring countries, especially Namibia, are also crucial to cheetah conservation. Namibia and Botswana protect approximately 77% of the southern African cheetah population (IUCN/SCC, 2007). According to the cheetah range map provided by Marker et al. (2007) for Namibia, the most important population linkage between the two countries lies in the Ghanzi Agricultural Zone. Our analysis gave the Ghanzi Farms stratum (stratum 7.1), which consists of commercial game and livestock farms, a rank of medium suitability for cheetahs (Fig. 5). This is in concurrence with Kent (2010) who found a reduced but healthy wild prey base, high human-cheetah conflict, and an estimated cheetah density of 0.61 and 1.30 cheetahs/100 km2. This is higher than the range of cheetah densities (0.1–0.35 cheetahs/100 km2) found on Namibian farmlands (Marker et al., 2007). In the Ghanzi Community stratum (stratum 7.2) to the south, which consists of communal grazing land, cheetah prey biomass was high but the percentage cheetah prey was low; human-cheetah conflict was thus predicted to be high. Its rank of low suitability is in concurrence with Muir (2009) who found cheetahs were rarely reported by farmers or wildlife officers in the Hanahai Tribal Grazing Area, which falls within this stratum. Conflict with humans, rather than prey availability thus seems to be the major factor affecting the survival of cheetahs in the Ghanzi Conservation Zone.

Conclusions

Botswana’s landscape provides large tracts of land that are suitable for the long-term-persistence cheetahs. However, our predictive map highlights areas within Botswana that warrant more investigation and intervention by conservationists. Those areas that we predict to be unsuitable for cheetahs, but are nonetheless important potential corridors for the species, should receive particular attention. In these areas, we recommend that wildlife corridors are maintained either on agricultural land with consultation with local farmers, or areas are set aside where farming activities are limited. Where farmers and cheetahs currently coexist in Botswana, we recommend that wild prey species be maintained at a level of at least 20% of the overall prey biomass as part of a comprehensive conflict mitigation strategy.

Supplemental Information

Data S1 Small livestock species predated on by cheetahs in Botswana

Livestock attacks were recorded in problem animal control (PAC) reports from the Department of Wildlife and National Parks (DWNP) (unpublished data).

Click here for additional data file.

Data S2 Cheetah prey, small stock, and percentage cheetah prey biomass (LSU) per 12′ aerial survey grid used in the analysis.

We calculated the mean number per cheetah prey species for each 12′ grid cell (approximately 21 × 22 km) for the 2001 to 2005 and 2007 aerial surveys. The mean number per species were then converted to species biomass in Large Stock Units (LSU), using the formula LSU = number of animals × body weight0.75 (Meissner, 1982), and summed to provide the total cheetah prey and small stock biomass, respectively, per grid cell. Percentage cheetah prey biomass is the proportion of cheetah prey biomass relative to the total biomass (cheetah prey and small stock combined).

Click here for additional data file.

Data S3 The predator management zones, stratum names, and identification numbers used in the figures

Based on the distribution of cheetah prey biomass, small stock biomass, and percentage cheetah prey biomass, we refined the predator management zones, identified in the broad landscape suitability stratification for large carnivores in Botswana from Winterbach, Winterbach & Somers (2014), into homogeneous strata.

Click here for additional data file.

Data S4 The number and percentage of aerial survey grids with ≤20% cheetah wild prey with corresponding suitability class

The percentage grid cells per stratum with ≤20% cheetah prey and the Natural Breaks (Jenks) function in ArcMap 9.3.1 to classify the strata into five classes of suitability were used in the predictive map of the landscape suitability in Botswana for cheetahs.

Click here for additional data file.

Thanks to Laurie Marker, JW McNutt, E Fabiano, R Kotze and G Potgieter for valuable comments on the manuscript. This manuscript was made possible by the Department of Wildlife and National Parks, Ministry of Environment, Wildlife and Tourism, Gaborone, Botswana.

Additional Information and Declarations

Competing Interests

Author Contributions

This study was supported by Tau Consultants (Pty) Ltd., a registered independent research company in Botswana, of which HEK Winterbach and CW Winterbach are directors and sole shareholders. There are no patents, products in development or marketed products to declare and does not alter our adherence to PeerJ policies on sharing data and materials. Lorraine Boast and Rebecca Klein are both employees of Cheetah Conservation Botswana, and Michael J. Somers is an Academic Editor for PeerJ.

Hanlie E.K. Winterbach conceived and designed the experiments, performed the experiments, analyzed the data, wrote the paper, prepared figures and/or tables.

Christiaan W. Winterbach conceived and designed the experiments, performed the experiments, analyzed the data, prepared figures and/or tables, reviewed drafts of the paper.

Lorraine K. Boast and Rebecca Klein contributed reagents/materials/analysis tools, reviewed drafts of the paper.

Michael J. Somers reviewed drafts of the paper.

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
