# Peer review of "Relative availability of natural prey versus livestock predicts landscape suitability for cheetahs Acinonyx jubatus in Botswana"

_PeerJ, doi:10.7717/peerj.1033_

## Round 0.1 · original submission · Minor Revisions

Dear Hanlie E. K. Winterbach,

Congratulations for your work. Both reviewers and myself liked your paper very much. We all think that it is a nice approach and a nice contribution to the discussion about large carnivore conservation.

You will find the comments and suggestions of both reviewers below. Follow them to improve the manuscript before the final acceptance of the paper for publication in PeerJ,

best regards,

Sara Varela

·

Basic reporting

The paper was well written with clear figures.

Experimental design

While not a pure experiment, it is a well designed ecological modelling exercise that clearly describes primary research with a clear research question that is relevant and highly meaningful to both Botswana and more widely wherever humans and carnivores coexist. The methods are well described (but see below for some minor improvements) and I have no ethical concerns.

Validity of the findings

The data seem very robust to me and the findings valid.

Additional comments

I liked this paper and think it makes a great contribution to help improve our understanding of human-carnivore conflict. Minor considerations to improve the paper are:

- Abstract Line 4: what are 'suitable conservation areas beyond protected area boundaries'?

- Abstract 2nd last sentence: 'Twenty percent of wild prey...'

- L8-9: '...can also strongly influence their movements...'

- L14: full stop needed.

- L16: '... to effectively conserve them both inside...'

- L23: you might want to cite Belbachir, F., N. Pettorelli, T. Wacher, A. Belbachir-Bazi & S. M. Durant (2015) Monitoring Rarity: The Critically Endangered Saharan Cheetah as a Flagship Species for a Threatened Ecosystem. PLoS ONE, 10, e0115136. as it gives a great illustration of how wide ranging cheetahs can be.

- L26-29: I think it would be worth further justifying this point by highlighting which protected areas actually support viable (Ne >500 or 4000 depending on who you follow) populations of cheetahs.

- L47: '...as cheetahs means we can ill...'

- L48: '... best available and reliable information...'

- L73: Isn't it Makgadikgadi/Nxai Pans National Park? In which case the acronym should change.

- L115: who is the pers comm from?

- L119-121: Isn't the % of wild prey out of all prey correlated with the 1st 2 variables? If so, you'll need to reanalyse this.

- L135: '...critical percentage of wild prey...'

- L137: more details on the questionnaire.

- L146: 'from expected'

- L175: The justification for having two categories of 20% and a 20-80% category is confusing.

- L198: '20% wild prey biomass was a potentially ...'

- L288 and elsewhere: I think 'wide-spread' is one word 'widespread'.

I hope this is helpful.

Yours sincerely

Matt Hayward

·

Basic reporting

Overall, this is a very interesting approach to a challenging conservation and management issue.

The Introduction provides a clear description of the state of the problem and why the authors have chosen to address the management of cheetahs using this alternative and less invasive approach.

There could have been a bit more detail on how the work fits into the broader field of knowledge with regard to the current status of knowledge on cheetahs in Botswana and the ways in which current management practices are and/or are not working. However, there was certainly a body of prior literature that was included in the background information provided.

Within the manuscript, there are areas where basic editing for style and grammar could benefit the narrative. Attention to these areas should improve the work and provide some clarity. Notes on specific areas for improvement are provided in a marked up copy of the manuscript.

Experimental design

The design of the research in this manuscript appears to be a unique but well thought out approach. Clearly, predator dynamics are driven by prey populations, and I have not seen prior work where habitat suitability was estimated solely based on prey and risk of human conflict. This is a novel idea for managing large predators in an environment with mixed habitats and land-use types. However, there were a number of things that were not clear from the work, as submitted.

The Introduction includes conclusions that should be reserved for the discussion and perhaps referenced in the Abstract. Furthermore, these statements are made at the end of the Introduction where there should be more attention to clearly and explicitly describing what the research question is and how you plan to get there. While the general gist of the study is provided, more detail would help solidify the remainder of the manuscript an would likely also help guide organization and structure.

There was insufficient or unclear information in the Methods and Results with regard to how the estimations of biomass were calculated. More description is needed on how species were identified and selected, how multiple surveys were combined to arrive at a single count, and how aerial surveys were translated into the LSUs described in the text. As it stands, it was difficult to keep track of how each parameter described was calculated and the results from each, let alone reproduce the findings of this work from the data. Additionally, the categorization of the different results and eventual habitat suitability modeling need to be explained much more clearly so the reader can follow or even reproduce results described. Perhaps subheadings for each step of the approach would help with organization and to ensure adequate description of each step in the analysis process.

The terms identified for each prey parameter need to be consistently used throughout. The organization of the Methods, Results, and Discussion should also follow the same pattern throughout for clarity.

More specifics on areas for improvement are provided in a marked up copy of the manuscript.

Validity of the findings

While the data this work was based on have been made available, it is difficult to assess whether all of the analyses are robust and statistically sound because the description of the approach needed to be clearer and contain more detail.

The conclusions presented in the Discussion appear to mostly be appropriate although there was extraneous information in this section that could be reduced and honed to more firmly support the key findings. This, in particular, is where I felt that having a more clearly and strongly stated research question and goals would have benefited the remainder of the manuscript. Focusing on fine scale details too early in the Discussion made it difficult to keep sight of the original goal and how well this approach helped with distinguishing suitable cheetah habitat and developing appropriate conservation targets to protect and/or improve cheetah habitat and populations.

Also, at some point in the Discussion, cheetah density is mentioned. If this type of data is available, it would be very interesting to use those data (even if not available for the entire study area) to attempt to validate the findings. While I recognize that the farmer surveys and damage/attack reports were an attempt to validate the suitability maps, utilizing data on cheetah densities, even if just estimates, would be interesting to compare for areas categorized as high vs. low using your proposed approach to quantifying habitat suitability.

Additional comments

I appreciated the opportunity to review this article and thought this was a truly novel approach to address management challenges faced by those working to conserve large carnivores and their habitat all over the world.

---

## Round 0.2 · accepted · Accept

Dear Hanlie Winterbach,

I am glad to tell you that your article has been accepted for publication in PeerJ